# Alcohol’s Impact on the Cardiovascular System

**DOI:** 10.3390/nu13103419

**Published:** 2021-09-28

**Authors:** Michael Roerecke

**Affiliations:** 1Centre for Addiction and Mental Health (CAMH), Institute for Mental Health Policy Research, 33 Ursula Franklin Street, Toronto, ON M5T 2S1, Canada; michael.roerecke@camh.ca; 2Campbell Family Mental Health Research Institute, CAMH, 33 Ursula Franklin Street, Toronto, ON M5T 2S1, Canada; 3Dalla Lana School of Public Health, University of Toronto, 155 College Street, Toronto, ON M5T 1P8, Canada

**Keywords:** alcohol drinking, binge drinking, cardiovascular diseases, ischaemic heart disease, hypertension, stroke, review

## Abstract

Alcohol consumption has been shown to have complex, and sometimes paradoxical, associations with cardiovascular diseases (CVDs). Several hundred epidemiological studies on this topic have been published in recent decades. In this narrative review, the epidemiological evidence will be examined for the associations between alcohol consumption, including average alcohol consumption, drinking patterns, and alcohol use disorders, and CVDs, including ischaemic heart disease, stroke, hypertension, atrial fibrillation, cardiomyopathy, and heart failure. Methodological shortcomings, such as exposure classification and measurement, reference groups, and confounding variables (measured or unmeasured) are discussed. Based on systematic reviews and meta-analyses, the evidence seems to indicate non-linear relationships with many CVDs. Large-scale longitudinal epidemiological studies with multiple detailed exposure and outcome measurements, and the extensive assessment of genetic and confounding variables, are necessary to elucidate these associations further. Conflicting associations depending on the exposure measurement and CVD outcome are hard to reconcile, and make clinical and public health recommendations difficult. Furthermore, the impact of alcohol on other health outcomes needs to be taken into account. For people who drink alcohol, the less alcohol consumed the better.

## 1. Introduction

Alcohol is one of the most important risk factors for disease and mortality globally [1]. The relationship between alcohol consumption and cardiovascular diseases (CVDs) is complex, and hundreds if not thousands of individual research reports have been published. Due to the potential beneficial effects of alcohol consumption on some CVD outcomes, the relationship between alcohol consumption and CVDs, in particular ischaemic heart disease (IHD), is controversial and highly debated [2,3,4,5,6,7].

Diseases under the umbrella of CVDs are differential in their aetiology. Therefore, based on major reviews and meta-analyses, this review is divided into the major CVD sub-categories: IHD, ischaemic stroke (IS), haemorrhagic stroke (HS), hypertension, atrial fibrillation (AF), cardiomyopathy, and heart failure.

Alcohol consumption is multi-dimensional, and there is no agreement in the literature on how to label different levels of alcohol consumption. What is considered low, moderate, and heavy alcohol consumption varies widely. Even the amount of alcohol in a ‘standard’ drink varies considerably [8]. For example, in the UK, one standard unit is 8 g pure alcohol (half a pint); in the US, it is 14 g per standard drink; and in Canada, it is 13.6 g. Episodic heavy drinking, sometimes called ‘binge’ drinking, is not consistently defined [9,10,11,12]. In the US, episodic heavy drinking occasions are defined as alcohol consumption that brings the blood alcohol concentration to at least 0.08% (or 0.08 g of alcohol per deciliter), corresponding to ≥5 US standard drinks per occasion in men and ≥4 standard drinks in women, in about 2 h [13]. Accordingly, in this narrative review, in order to standardize the exposure, alcohol intake is referred to in grams of pure alcohol based on reported conversion factors. The following search terms were used in Medline and Embase: (cardiovascular diseases or cardiac diseases or stroke or heart diseases or heart failure or cardiac myopathy or cardiac arrhythmia or hypertensive heart disease or hypertension or high blood pressure or elevated blood pressure or resistant hypertension).mp AND (alcohol consumption.mp. or exp alcohol consumption) AND (systematic reviews and meta-analysis).mp.

## 2. Ischaemic Heart Disease

Alcohol consumption and IHD are both highly prevalent in high-income countries. Many systematic reviews and meta-analyses [5,14,15,16,17,18,19,20] and numerous individual studies have been published in recent decades on the relationship between alcohol consumption and IHD, or myocardial infarction, the main subcategory of IHD. This relationship and its implications remain controversial due to a lack of long-term randomized controlled trials with CVD endpoints. Most meta-analyses of epidemiological data on the topic have found a J-shaped or sometimes inverse relationship between average alcohol consumption and IHD, with lifetime abstainers showing a higher risk compared to current ‘moderate’ drinkers (various amounts of alcohol are used to define these drinking groups), and then an uptake of the risk curve to similar or higher risks compared to those seen for heavier drinkers. Oftentimes, whether or not a J-curve or an inverse or U-shaped relationship is observed depends on the range of alcohol consumption reported in an individual study and the specific IHD endpoint considered (fatal or non-fatal). 

The J-shaped risk relationship has been found in both sexes and for IHD morbidity and mortality [16,21]. In a meta-analysis comprising 957,684 participants and 38,627 events, a J-shaped curve in relation to lifetime abstainers was observed in women for both fatal and non-fatal IHD outcomes, and an inverse relationship was observed in men with non-fatal IHD events [16]. Using only studies fully stratified by sex and endpoint, the nadir was found at 32 g per day for IHD mortality in men, 69 g per day for IHD morbidity in men, 11 g per day for IHD mortality in women, and 14 g per day for IHD morbidity in women. The evidence suggests that the type of alcoholic beverage does not play a role in the shape of the relationship. A meta-analysis [22] of fatal or non-fatal CVD events showed that a J-shaped association was observed for the consumption of wine, an inverse relationship for beer consumption, and a negative association for spirits.

IHD mortality among men who drink 60 or more g of pure alcohol on average per day has been found to be similar to that of lifetime abstainers [18]. Among women, such drinking levels are rarely observed in typical epidemiological studies. The risk from alcohol consumption is typically higher in women for the same amount of alcohol consumption compared to men, due to body fat distribution, body size, and alcohol solubility [23,24,25]. Nevertheless, both men and women with alcohol use disorders, who oftentimes, but not always, drink very heavily, have been associated with some of the highest mortality risks for IHD (RR = 1.62; 95% CI: 1.34 to 1.95 in men; RR = 2.09; 95% CI: 1.28 to 3.41 in women compared to the general population) [17]. 

Due to the large heterogeneity observed in meta-analyses of alcohol consumption and CVD outcomes, it is clear that not all drinking is associated with a lower risk for IHD. For example, data from Russia consistently show a detrimental association with IHD outcomes; however, it should be noted that the most prevalent drinking pattern in Russia at the time of these studies was infrequent heavy drinking rather than low amounts more frequently. Both drinking patterns would result in the same magnitude of average alcohol intake over the week [26,27]. 

Perhaps the most compelling observational evidence for a beneficial association between average alcohol consumption and IHD comes from an individual-participant analysis of nearly 600,000 current drinkers of the Emerging Risk Factors Collaboration, EPIC-CVD, and the UK Biobank cohorts [28]. In an inverse relationship, the hazard ratio for an increase in 100 g pure alcohol per week in comparison to >0 to <50 g per week was 0.94 (95% CI: 0.91–0.97) for myocardial infarction (14,539 events). However, except for myocardial infarction, the risk (per 100 g per week increase in consumption) for stroke (HR = 1.14, 1.10–1.17), coronary disease other than myocardial infarction (1.06, 1.00–1.11), heart failure (1.09, 1.03–1.15), and fatal hypertensive disease (1.24, 1.15–1.33) increased in a linear fashion. The shape of the relationship between average alcohol consumption and myocardial infarction was J-shaped for fatal myocardial infarction (2748 events) and non-fatal coronary disease excluding myocardial infarction (6000 events), and inverse for non-fatal myocardial infarction (11,706 events). The data point for the highest consumption was 300 g per week, which translates to about 25 standard drinks of 12 g each per week, or 3.57 standard drinks per day on average. The analyses were adjusted for age, sex, smoking, and history of diabetes. Due to the strong association with MI, the nadir (i.e., the lowest risk) for overall CVD events (39,018 events) was at an alcohol consumption level of 100 g per week.

The threat of unmeasured confounding variables and other sources of bias [7] is not unique to the alcohol–IHD relationship. Early on, the sick-quitter hypothesis [29] was widely thought to be the cause of the J-shaped curve reported in many studies. Meta-analyses have shown that former drinkers are associated with a higher risk for IHD mortality than lifetime or long-term abstainers (RR = 1.25; 95% CI: 1.15–1.36) [30]. No association was found for both sexes for IHD morbidity. The sick-quitter hypothesis has been systematically evaluated, and meta-analyses have shown that even when lifetime abstainers are the reference group, thereby eliminating the sick-quitter effect, or, in other words, former drinking bias, a lower risk for people reporting alcohol consumption up to 30 g per day without irregular heavy drinking episodes remained [5].

Aside from the issue of reference groups, heavy episodic drinking, i.e., drinking about five standard drinks for men and four for women on one occasion or within two hours based on some definitions [13], seems to be an effect modifier for the relationship between average alcohol consumption and IHD. A meta-analysis [5] found an RR = 1.75 (95% CI: 1.36–2.25) for people who drink up to 30 g on average per day, but who actually have a drinking pattern characterized by less frequent drinking and mostly heavy drinking episodes compared to drinkers without such a drinking pattern. This increased risk seems to negate any lower risk for IHD found in people who drink up to 30 g on average per day without heavy drinking episodes. Thus, the risk was similar for lifetime abstainers and people who consume alcohol mostly in heavy drinking episodes. However, the concept of heavy episodic drinking, at least regarding the CVD effects of alcohol consumption, is not clearly defined, and different studies use different thresholds for heavy episodic drinking. These vary from three drinks per drinking day to six or even eight drinks per drinking day [31,32]. For an overview of mechanisms, please see [33,34].

Adjustment for possible confounders, some of which may lie in the pathway of CVD development and could be considered mediators, remains an issue in alcohol epidemiology [35]. An analysis of individual-level data from eight cohort studies showed that adjustment for age; year of baseline; smoking; body mass index; education; physical activity; energy intake; intake of polyunsaturated fat, monounsaturated fat, saturated fat, fiber, and cholesterol; and study design did not explain the J-shaped association [36]. 

A J-shaped or inverse association has also been reported in patients with CVD. In a meta-analysis of 11 cohorts published in 2014, an inverse risk relationship between average alcohol consumption and IHD in patients with hypertension was reported [37]. Similar associations have been reported among people with diabetes and non-fatal myocardial infarction [38,39,40,41,42]. A recent large-scale study from the UK reported a J-curve for most CVD outcomes in patients with CVD [43].

The pathways by which alcohol consumption may exert a beneficial effect on ischaemic diseases are not well understood. In the absence of long-term randomized controlled trials on CVD endpoints, 44 intervention studies on surrogate biomarkers for CVD were summarized in a meta-analysis in 2011 [44]. The results showed a substantial dose–response relationship for high-density lipoprotein-C with (in comparison to no alcohol consumption): mean difference: 0.072 mmol per L (95% CI: 0.024–0.119) for 12.5–29.9 g per day; mean difference: 0.103 mmol per L (95% CI: 0.065–0.141) for 30–60 g per day; mean difference: 0.141 mmol per L (95% CI: 0.042–0.240) for >60 g per day. The effect on fibrinogen levels was −0.20 g per L (95% CI: −0.29 to −0.11), and for Adiponectin, 0.56 mg per L (95% CI: 0.39–0.72). Alcohol consumption did not substantially change the levels of total cholesterol, low density lipoprotein cholesterol, triglycerides, Lp(a) lipoprotein, C-reactive protein, interleukin 6, or tumour necrosis factor α. Analyses stratified by type of alcoholic beverage were similar to analyses of all alcoholic beverages combined. 

Due to the limitations of typical epidemiological studies, other types of study design, such as Mendelian randomization studies using an instrumental variable approach, sought to answer questions about the causality of the lower risk of low-level alcohol drinkers. However, the use of such an approach [45,46], which depends on several assumptions that are not easily met in a complex relationship, such as between alcohol consumption patterns and CVD risk, is highly debated [47,48,49,50]. 

## 3. Hypertension

Several meta-analyses have been published over the last two decades that summarize the relationship between average alcohol consumption and incidence of hypertension [51,52,54,55,56,57,58]. While older reviews (e.g., [54,58]) found a small but significantly lower risk in women who reported very small amounts of alcohol intake, more recent meta-analyses with more data did not find such an association. In particular, in a meta-analysis of 361,254 participants from 20 cohort studies (125,907 men and 235,347 women) with 90,160 incident cases of hypertension, the risk compared to abstainers was elevated for any amount of alcohol consumption in men, and in women, there was no increased risk for up to 24 g per day (RR = 0.94; 95% CI: 0.88–1.01), with an increased risk beyond this level of consumption [57]. The risk to former drinkers was similar to lifetime abstainers (RR = 1.03; 95% CI: 0.89–1.20). Among men, the risk increased to 1.68 (95% CI: 1.31–2.14) for drinking 60 g per day on average. No such data were available for women. The difference in risk for up to 24 g per day for women compared to men was significant (RR = 0.79; 95% CI: 0.67–0.93). One possible explanation for this difference could be more detrimental drinking patterns among men, which typically includes more binge drinking episodes. Heavy episodic drinking elevates blood pressure and, subsequentially, the risk for hypertension [73,74]. 

The relationship between alcohol consumption and blood pressure and hypertension has to be seen as causal and reversible, with experimental evidence showing that a reduction in alcohol consumption leads to a reduction in both systolic and diastolic blood pressure in a dose–response relationship with effects of clinical importance [75]. The reduction in systolic blood pressure is sizable with a mean difference of −5.50 mmHg (95% CI: −6.70 to −4.30) for people who drink 72 g per day on average and reduce their consumption by about 50%. There was no discernible difference for drinkers of up to 24 g per day in comparison to abstainers; however, data were sparse. For a discussion of mechanisms, please see [74,76].

## 4. Stroke

There are two major stroke subtypes with differing aetiologies: IS (based on ischaemic disease processes) and HS (based on haemorrhagic processes, i.e., bleeding processes). Due to higher prevalence, IS typically drives investigations of total stroke. With similarities in aetiology, one would expect IS to show a similar relationship with alcohol consumption in comparison to IHD. Indeed, several earlier [52,60] and more recent [59,62] meta-analyses have shown that the association between average alcohol consumption and IS follows a J-curve. In a meta-analysis of 27 prospective cohort studies with 3824 IS cases (2216 men and 1608 women) compared to abstainers, the risk for IS was below one for up to 24 g per day on average (RR = 0.90 (95% CI: 0.85–0.95) for <12 g; RR = 0.92 (95% CI: 0.87–0.97) for 12–24 g per day), and increased for alcohol consumption >24 g per day [59]. In contrast, the analysis of the Emerging Risk Factors Collaboration, EPIC-CVD, and the UK Biobank cohorts [28] showed an increased risk for both fatal and non-fatal total stroke based on average alcohol consumption.

The risk for intracerebral and subarachnoid HS increased with every drink, and the consumption of >48 g per day resulted in an RR = 1.67 (95% CI: 1.25–2.23) for intracerebral stroke and 1.82 (95% CI: 1.18–2.82) for subarachnoid HS [59,61].

Several studies have reported an elevated risk for both IS and HS from heavy episodic drinking [77,78,79]. One study showed that the risk increased with a higher frequency of heavy episodic drinking [78]. Alcohol consumption is also a trigger for stroke events. The higher the alcohol consumption within 24 h or one week, the higher the risk for IS or HS [53,80].

## 5. AF and Cardiomyopathy

Several meta-analyses have investigated the risk of AF in relation to alcohol consumption [63,64,65,66]. In a meta-analysis of seven cohort studies with 12,554 cases of AF, in comparison to non-drinkers, the risk for AF was elevated in all drinking groups, even when heavy episodic drinkers were excluded from the analysis. The pooled linear risk increase was 1.08 (95% CI: 1.06–1.10) for each 12 g per day increase in average alcohol consumption. More recently, using data from 249,496 participants, a meta-analysis concluded that there was no increase in risk for AF for consumption of 6–7 drinks (10–12 g per drink) per week [63]. Another large cohort study of 403,281 participants from the UK Biobank with 21,312 incident cases of AF reported a J-shaped relationship for average alcohol consumption, with people drinking 56 g per week or less having the lowest risk in comparison to lifetime abstainers [67]. A beverage-specific analysis showed that the J-shaped curve was found in wine drinkers, but not in beer or spirit drinkers [67]. A recent randomized controlled trial indicates that a reduction in drinking is associated with a lower recurrence of AF [81].

While the exact amounts remain unknown, alcohol consumption (either regular or irregular), can cause, aside from hypertension, structural damage to the heart muscle and arrythmias [68]. Cardiomyopathy, which is characterized by ventricle dilatation, hypertrophy, and dysfunction, can be caused by alcohol consumption and its metabolites, both of which have a direct toxic effect on the heart muscles [68]. A quantification of a potential dose–response relationship has not been possible to date; however, it seems that the consumption of >80 g per day leads to a substantially increased risk [69]. Few studies have been conducted among women, for which less alcohol consumption per day and a short duration of such consumption over several years have a similar effect compared to men. It has been estimated that 1–40% of alcohol use disorder patients have cardiomyopathy, or, conversely, that 23–47% of patients with dilated cardiomyopathy have, in fact, alcoholic cardiomyopathy [69].

## 6. Heart Failure

CVD categories, such as IHD, hypertension, and cardiomyopathy, increase the risk of heart failure. Three meta-analyses largely came to similar conclusions [70,71,72]. In the most recent meta-analysis published in 2018, which consisted of 355,804 participants with 13,738 cases of HF based on 13 cohort studies, it was shown that the dose–response relationship between alcohol consumption and HF was curvilinear [71]. Compared to non-drinkers, the risk for 1–84 g per week, 85–168 g per week, 168–336 g per week, and >336 g per week were RR = 0.86 (95% CI: 0.81–0.90), 0.88 (0.77–1.01), 0.91 (0.80–1.04), and 1.16 (0.92–1.47), respectively. Based on eight studies, the meta-analysis concluded that former drinkers are at a higher risk for AF compared to lifetime abstainers (RR = 1.22; 95% CI: 1.11–1.33) [71]. Due to data limitations, the role of sex and other potential effect modifiers remains unclear.

## 7. Conclusions

Epidemiological studies indicate a complex relationship between various dimensions of alcohol consumption (i.e., life course drinking patterns) and CVD outcomes. Indeed, substantial heterogeneity is evident. Most epidemiological studies to date have relied on a single measurement of alcohol intake at baseline. It is assumed that the self-reported drinking levels, preferably including drinking patterns, remains the same before and after the baseline measurement. For many people this is clearly not the case, and even lifetime abstainers are hard to identify [82]. 

Does some alcohol consumption protect some people against ischaemic diseases to some degree? Epidemiological data, as outlined in this review, suggest that this is the case (Table 1). For example, a J-shaped relationship emerges for average alcohol consumption and IHD and IS. On the other hand, the relationship with incident hypertension, which is a potent risk factor for most if not all CVDs, is quite different between men and women, with an increased risk for any amount of alcohol consumption in men. While potential sources of bias, such as the reference group, i.e., separating lifetime abstainers, former drinkers, and heavy episodic drinkers, have been systematically investigated for the relationship between alcohol and IHD, their impact on other CVD outcomes remains less clear. While there is a lack of large-scale randomized studies on the long-term effect of alcohol consumption on various CVD endpoints, short-term clinical trial data indicate a sizable effect of alcohol consumption on HDL-C and fibrinogen. However, the heterogeneity found in epidemiological studies points to more than just biological differences. Socioeconomic status, for example, might influence the impact of alcohol on CVD [83]. More research is necessary to advance knowledge on this topic.

It should also be noted that due to the limitations of alcohol-epidemiological studies, the beneficial associations tend to be overestimated. Furthermore, potential beneficial effects of non-heavy alcohol consumption on CVD endpoints, as described in this review, have already been observed at very low levels, such as 100 g pure alcohol per week, which, at the lower end, translates to about 1 drink every other day. As such, most drinkers should drink less. Recommending drinking as a primary or secondary prevention measure for CVDs, which comes up occasionally in the literature, should be discouraged due to the substantial risks of any alcohol consumption for many health outcomes. Alcohol is a carcinogen, neuro-toxin, hepato-toxin, and psychoactive drug. 

## Figures and Tables

**Table 1 nutrients-13-03419-t001:** Shape of the relation between alcohol consumption and CVD categories based on current evidence syntheses.

CVD Category	Shape of the Relationship	Meta-Analyses and Systematic Reviews
Ischaemic heart disease	J-shaped, modified by episodic heavy drinking	[5,14,15,16,17,18,19,20,21,22,28,30,37,43,44,51,52,53]
Hypertension	Any consumption detrimental in men, detrimental beyond 24 g in women; reduction in drinkers >24 g/day lowers blood pressure	[51,52,54,55,56,57,58]
Stroke	Ischaemic stroke: J-shaped, possibly modified by episodic heavy drinkingHaemorrhagic stroke: detrimental, possibly modified by episodic heavy drinking	[21,28,51,52,59,60,61,62]
Atrial fibrillation	Detrimental beyond 60 g/week	[63,64,65,66,67]
Cardiomyopathy	Consumption of >80 g/day leads to a substantially increased risk	[68,69]
Heart failure	J-shaped	[63,70,71,72]

## Data Availability

Not applicable.

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
