# Peer review of "Alcohol’s Impact on the Cardiovascular System"

_nutrients, 2021, doi:10.3390/nu13103419_

Round 1

Reviewer 1 Report

This narrative review discussed alcohol’s impact on the cardiovascular system.

This manuscript focused on the relation between pure alcohol intake and cardiovascular disease, and contains practical epidemiological data. Fundamentally, this is a well-written review. However, there are some issues that need to be addressed.

  1. Although the author reported that there is no agreement in the literature on how to label different levels of alcohol consumption, some pronouns were still used in this article to denote drinking levels, such as “very high alcohol consumption”, “binge drink”, “standard drinks”, etc. It would be helpful to the readers if the author can offer a list or a table for common definitions of these pronouns.
  2. Making tables in every section that include important studies and summarize their daily/weekly pure alcohol intake and the cardiovascular outcome (RR, OR or HR) may be helpful to the readers.
  3. In the “Atrial fibrillation and Cardiomyopathy” section, it may be worthy to cite the randomized controlled trial (N Engl J Med. 2020 Jan 2;382(1):20-28.), "Alcohol Abstinence in Drinkers with Atrial Fibrillation."
  4. Please check the location of square brackets for reference numbers. It should be placed before the punctuation marks.

Author Response

1. A list of standard drink definitions has been added. Binge drinking is now consistently described as episodic heavy drinking, with a definition.

2. A table summarizing the shape of the relation between alcohol consumption and CVD outcomes has been added (Table 1).

3. The reference has been added to the manuscript.

4. The reference markers have been revised.

Reviewer 2 Report

In this review Roerecke examines the epidemiological evidence for the associations between alcohol consumption, including average alcohol consumption, drinking patterns, and alcohol use disorders, on CVDs

This is an interesting paper summarizing the most important works on the topic performed by an expert in this field (considering also the self-references reported in the paper).

Minor comments:

  1. Please add the search procedures (medical subject heading, selection criteria, etc.. ) for this narrative review.
  2. A table summarizing the most important findings for each CVD should be added.
  3. CVD without s also in the abstract
  4. Line 15: hypertension, atrial…
  5. Line18-19: please rephrase.
  6. Line 205….risk of AF in relation to alcohol consumption.

Author Response

1. The search terms have been added.

2. A table summarizing the shape of the relation between alcohol consumption and CVD outcomes has been added (Table 1).

3. The manuscript has been revised in the suggested way.

4. The manuscript has been revised in the suggested way.

5. The manuscript has been revised in the suggested way.

6. The manuscript has been revised in the suggested way.